# Repeat Glacier Collapses and Surges in the Amney Machen Mountain Range, Tibet, Possibly Triggered by a Developing Rock-Slope Instability

**Frank Paul**

Department of Geography, University of Zurich, 8057 Zurich, Switzerland; frank.paul@geo.uzh.ch;
Tel.: +41-44-6355175

**Abstract:** Collapsing valley glaciers leaving their bed to rush down a flat hill slope at the speed of a racing car are so far rare events. They have only been reported for the Kolkaglacier (Caucasus) in 2002 and the two glaciers in the Aru mountain range (Tibet) that failed in 2016. Both events have been studied in detail using satellite data and modeling to learn more about the reasons for and processes related to such events. This study reports about a series of so far undocumented glacier collapses that occurred in the Amney Machen mountain range (eastern Tibet) in 2004, 2007, and 2016. All three collapses were associated with a glacier surge, but from 1987 to 1995, the glacier surged without collapsing. The later surges and collapses were likely triggered by a progressing slope instability that released large amounts of ice and rock to the lower glacier tongue, distorting its dynamic stability. The surges and collapses might continue in the future as more ice and rock is available to fall on the glacier. It has been speculated that the development is a direct response to regional temperature increase that destabilized the surrounding hanging glaciers. However, the specific properties of the steep rock slopes and the glacier bed might also have played a role.

**Keywords:** glacier surge; glacier collapse; rock-slope instability; hazard; Landsat; Sentinel 2; Tibet

## 1. Introduction

Glacier surges have recently been in the focus of several scientific studies [1], largely because remote sensing data with a sufficiently high spatial and temporal resolution allow accurate tracking of surface and morphological changes as well as creation of dense time-series of flow velocities [2–12]. During a surge, large amounts of ice are transported at comparably high velocities (several m/day) from an upper reservoir area downward to a receiving zone, possibly creating a strong advance of the terminus. After a surge, the ice becomes stagnant and melts away while in the reservoir zone new ice accumulates [8]. Whereas an improved understanding of possible surge mechanisms slowly emerges [13–15], collapses of glaciers (where the ice is removed from a flat bed) have so far only rarely been reported. The two most prominent and best-documented examples are the 2002 collapse of Kolkaglacier in the Caucasus [16–19] and the 2016 collapse of two small valley glaciers in the Aru mountain range of Tibet [20–23]. In both regions the collapses caused long-distance (several km) mass movements with very high velocities (200 to 290 km/h) [19,22] that have not been thought possible before they occurred. In contrast to the frequent ice break-offs at steep hanging glaciers [24], the Kolka and Aru glaciers rested on beds of a comparably low slope and should thus have been resistant to mechanical failure.

The history of events leading to the failure was different in both cases and they are thus unique on their own. Kolkaglacier is a well-known surge-type glacier [17] that is mostly nourished by avalanche snow and has been removed from its bed during its 2002 collapse, possibly due to a massive

loading with debris from rock fall of its surrounding slopes [25]. Kolkaglacier had a regular surge in 1969/70 and might have also collapsed in 1835 and 1902 [17]. The Aru glaciers were not known as surging before [23], but showed a typical pattern of surge-related elevation changes before they collapsed [21]. The twin-glacier collapse removed the lower parts of both glaciers, likely due to a change in thermal/hydrologic conditions at the glacier bed [22].

The collapses reported here for a glacier (GLIMS ID G099443E34824N, Inventory ID CN5J352E0017) located on the north-western slope of the Amney Machen (A'nyê Maqên) mountain range (34.822° N, 99.44° E) took already place in 2004, and again in 2007 and 2016. Figure 1a shows the location of the mountain range and the glacier and Figure 1b is an oblique perspective view up-glacier showing the study region after the second collapse in 2008. The surges and collapses of this glacier have so far not been reported [26] or analyzed in the scientific literature, although they repeatedly buried a country road that has been reconstructed after each event (Figure A1a). Hence, local authorities are aware of the collapses and an information board describing the 2004 avalanche had been installed. The board can be seen in a picture from a tourist [27] and shows the advancing glacier a few weeks before its third collapse in 2016 (Figure A1b). The small valley glacier (size about 1.5 km$^2$) has a homogenous, gently sloping tongue (slope about 13°) with a length of about 2 km and an elevation range of 450 m (from 4800 to 5250 m). The tongue is nourished by ice from a steep (mean slope 36°) and its once connected upper part, that reaches as high as 5900 m.

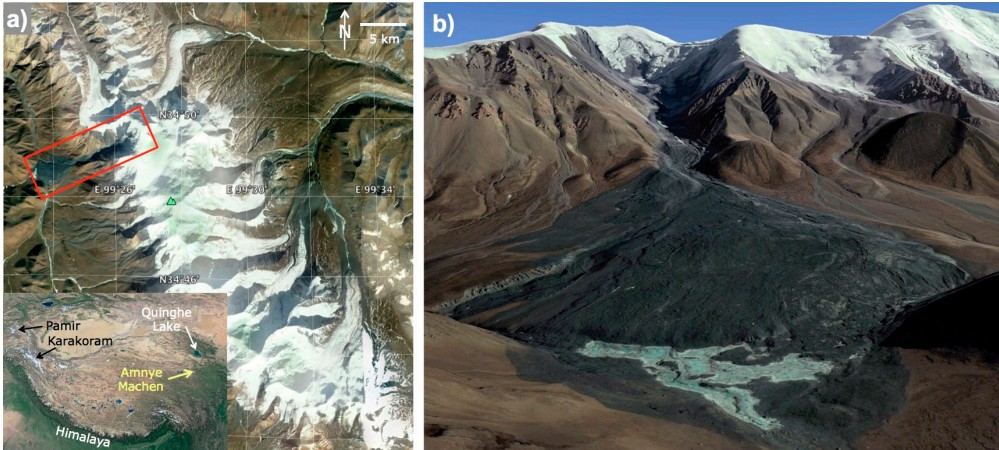

**Figure 1.** (**a**) Location of the Amney Machen mountain range in north-eastern Tibet, China. The collapsed glacier is marked with a red square. The location of the study region is marked in the inset with a yellow arrow. Image sources: Screenshots from Google Earth. (**b**) Oblique perspective view of the collapsed glacier as seen on a Quickbird image acquired on 13.1.2008, a few months after the second collapse. The steep and partly already glacier-free rock walls can be seen in the background. The extent of the 2004 collapse is still visible in the foreground, the deposit from the 2007 collapse can be seen on top of it. Image source: Screenshot from Google Earth.

The Amney Machen mountain range is covered by numerous glaciers (covering about 80 km$^2$), of which several have been classified in a previous study as surge type and strongly advancing between 1966 and 1981 [28]. Time series of Landsat images (available since 1987) reveal that several of the larger glaciers in this mountain range have surged again during the past three decades. Without temperature measurements, any assumptions about thermal conditions of the ice and rock walls are speculative. However, thermokarst features that can be found in the valley floor to the southwest of the mountain range and the permafrost zonation map by Gruber (2012) [29] are providing evidence that the glaciers higher up could be poly-thermal or cold-based. The purpose of this study is providing an overview on the glacier surges and collapses between 1987 and 2017 based on the analysis of freely available satellite image time series. A further analysis of the events using numerical modeling and field data is worthwhile but beyond the scope of this study.

## 2. Datasets

### 2.1. Satellite Data

The analysis of the surges and collapses of the glacier is based on optical satellite images from different sources (Table A1). Declassified reconnaissance imagery from the Corona KH4-A and B missions acquired in 1964 and 1969 are among the earliest sources of information about the region. With a spatial resolution of about 3 to 5 m they reveal several details of the glacier tongue and its forefield. However, sun-lit snow is over exposed. Landsat imagery covering the period 1987 to 2016 are mostly taken from Landsat 5 (21 scenes) at 30 m resolution (red-band), and Landsat 7 (14 scenes) at 15 m resolution (panchromatic band). The striping of Landsat scenes after 2003 due to the failure of the scan-line corrector had little impact on this, as the study region is located close to the scene center. Four Landsat 8 scenes acquired from 2013 to 2016 (15 m resolution) and four Sentinel 2 scenes acquired in 2016 and 2017 (10 m resolution) have been used for the more recent analysis. Collectively, satellite data provide an image in about every year for the full period (none in 1992/98), allowing a continuous interpretation of events. Several further scenes have been used for a detailed analysis of the collapses, partly also including scenes with snow cover and clouds. Key images of the surges and collapses are provided as a separate dataset in the Supplemental Material.

Additional analysis was performed using very high-resolution imagery acquired by Quickbird on 13 January 2008 and likely one of the Worldview satellites in winter 2016/17. These images were directly analyzed at maximum resolution in Google Earth and maps from bing.com. For the latter, the image source is not provided. The Corona and Landsat 5/7 scenes have been downloaded from earthexplorer.usgs.gov. The Landsat 8 and Sentinel 2 images were obtained from remotepixel.ca.

### 2.2. Digital Elevation Models (DEMs)

For topographic analysis and calculation of elevation changes, the 30 m version of the SRTM DEM (acquired in February 2000) has been used in combination with the 8 m High Mountain Asia (HMA) DEM acquired in 2015 [30]. Hillshade versions of both DEMs are shown for the study region in Figure 2. The latter suffers locally from data voids but is otherwise of very high quality. The SRTM DEM has a more rough or 'bumpy' surface but is otherwise looking reasonable. The SRTM DEM was downloaded from earthexplorer.usgs.gov and the HMA DEM from nsidc.org.

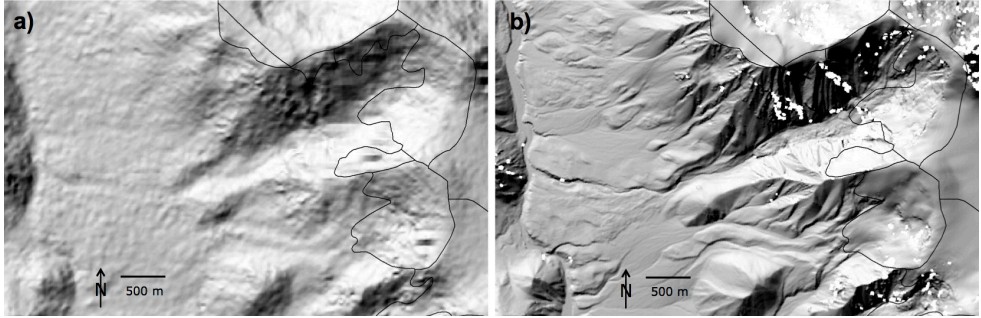

**Figure 2.** Visual comparison of the two DEMs for the region around the collapsed glacier (outlines from RGI6.0 in black). (**a**) The 30 m SRTM DEM from 2000 (source: earthexplorer.usgs.gov), (**b**) the 8 m HMA DEM from 2015 (source: nsidc.org/data/HMA_DEM8m_CT).

## 3. Methods

### 3.1. Satellite Data

As a base for quantitative assessments, glacier outlines were mapped automatically from the Sentinel 2 scene acquired on 4 August 2017 using the red/SWIR band ratio method [31]. The various extents of the glacier, the three debris fans, and the lake were created by on-screen digitizing using

ArcGIS from ESRI. Contrast-enhanced versions of the highest resolution dataset (panchromatic bands from Landsat 7 and 8) are used for this purpose. To identify the chronology of the surge and collapse events, all images were displayed in chronological order in a Geographic Information System (GIS) and flicker-images (going back and forth between two scenes) are used to follow the changes described below. The changes are difficult to see in side-by-side comparisons of static images, so some of the multi-panel images have been arranged to facilitate top-bottom comparison. For improved visibility of the changes the reader is referred to the Supplemental Material.

*3.2. Topographic Analysis*

The HMA DEM provided elevation and slope values of the glacier and the surrounding mountain range. In combination with the digitized glacier extents and deposit regions, elevation ranges have been calculated to determine mean slope values. The related horizontal distances were directly measured in the GIS using the Sentinel 2 image from 2017 in the background, i.e., the geometric reference has UTM zone 47N with WGS 1984 datum. Elevation changes were calculated by subtracting the SRTM DEM from the HMA DEM, after both were resampled bilinearly to 16 m resolution and co-registered. Horizontal shifts were smaller than the cell size, but the different vertical datum caused a bias of about 30 m that was subtracted. Due to artifacts in the difference DEM, elevation changes were only analyzed locally. A manually digitized centerline through the glacier and deposit area was used to derive elevation values from the co-registered DEMs.

## 4. Results: The History of the Surge and Collapse Events

*4.1. The 1960s Situation*

The Corona images from 1964 (Figure 3) and 1969 (not shown) revealed that the glacier was connected to its eastern and southern tributary during that time and had a well developed tongue with some debris cover along its northern margin. In the 1969 image, a crevasse across its entire width is visible in its flat upper part (indicating a steep slope in the bedrock) and a small rock outcrop (in the following RO-A) already existed in the northern part of its accumulation region. The terminus was flat and retreating from 1964 to 1969 by about 50 m. The river leaving the glacier has eroded an increasingly deeper trench in the foothill section of the mountain slope before reaching the main river (Qu'ngoin He) draining the region into the Yellow River (Huang He). In its lower part, the outflow is thus spatially well constrained, whereas in its upper part, erosion was less intense and a potential flooding would have been able to leave the riverbed and spread out over a larger region. Apart from this, nothing unusual (or a previous surge) can be detected on the images. The hill-slope section below the glacier seems to be intact, i.e., not impacted by a previous collapse.

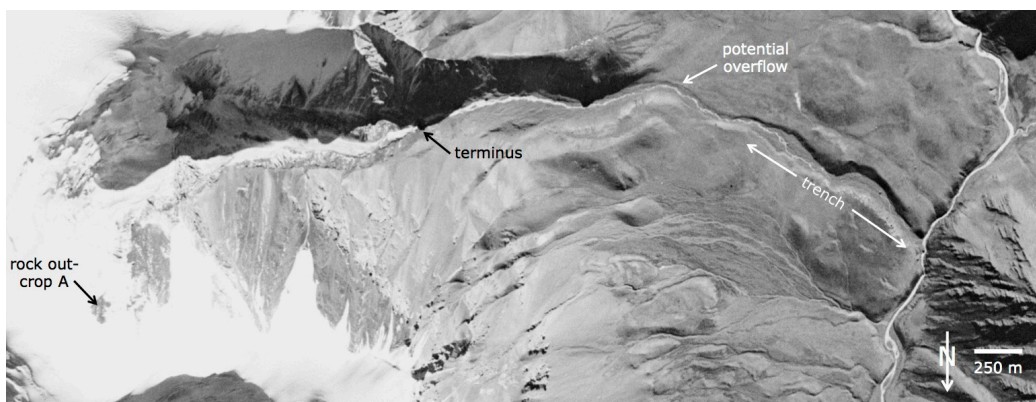

**Figure 3.** Corona image from 1964 showing the glacier being connected to its accumulation area and a glacier forefield that is free of any avalanche deposits. Note: South is up to highlight the relief. Image source: earthexplorer.usgs.gov.

### 4.2. The 1987–1995 Surge

The few available Landsat Multispectral Scanner images covering the period 1972 to 1982 are too sparse and too coarse to reveal any substantial variability in terminus position. It is thus assumed here that the glacier had been about stagnant during the 1970s until 1985. The following Landsat TM time series reveals a surging glacier tongue that advanced about 700 m from 1987 to 1995 (Figure 4). The size of RO-A increased during this time and by 1997 it connected to the ice-free terrain further down and was thus no longer an outcrop. As it is unlikely that all the ice has been removed by melting (other glaciers at lower elevations are stable), it can be assumed that a larger part of it was deposited on the lower glacier in form of ice avalanches. From 1996 to 2000, the terminus had been stationary and the lower part of the glacier tongue showed the typical post-surge down-wasting.

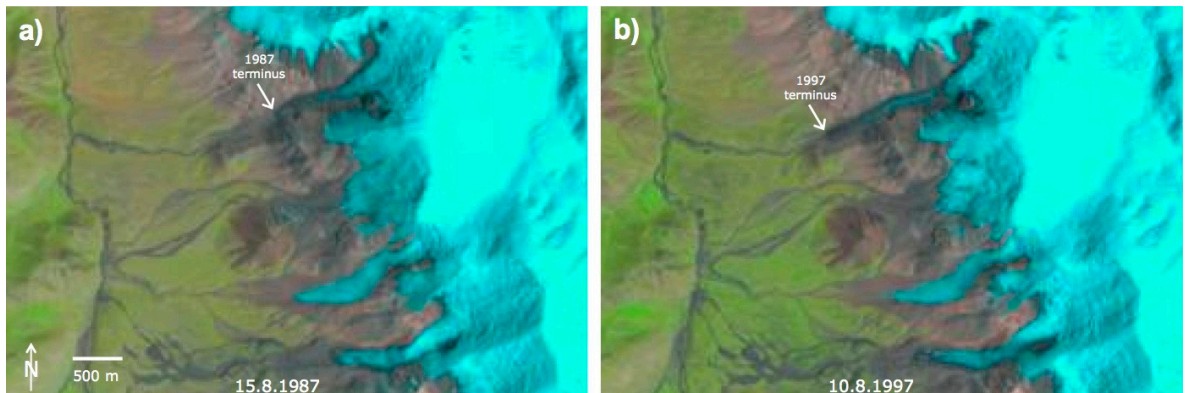

**Figure 4.** Landsat images from (**a**) 15 August 1987 and (**b**) 10 August 1997 showing the first surge of the glacier. The false colour images are showing (clean) ice and snow in cyan, rocks and gravel in pink to purple, and vegetation in light green. Image source: earthexplorer.usgs.gov.

### 4.3. The Developing Slope Instability

From August 2000 to August 2002, the area of the ice-free terrain around the former RO-A further increased, particularly in its upper parts (Figure 5a,b). The 2002 image also shows substantial darkening of the northern part of the glacier (in its flat section), indicating that substantial rock fall occurred before, maybe as a part of ice avalanches, maybe independently. This debris-covered part remained until 2003. From 2002 to 2003, the lower part of the glacier did not change much but a further rock-outcrop (RO-B) developed in the steep part of its eastern accumulation region. When also considering the later very high-resolution satellite images, it seems that between August and September 2003, a larger part of the main eastern tributary broke off and slid down onto the lower part of the glacier. Afterwards, the terminus started surging.

### 4.4. The 2003/4 Surge and Collapse

The glacier advanced by about 230 m and collapsed at some point between 26.1.2004 (Figure 5c) and 3.2.2004 as revealed by a largely snow-covered Landsat TM image. The ETM+ pan image form 11.2.2004 shown in Figure 5d has already much less snow so that the debris fan becomes visible. The bright trace in the panchromatic band of the ETM+ image and the still snow covered part visible on the TM image is indicating that a larger part of the collapsing glacier tongue has been deflected to the south by a double moraine wall at the end of the short valley and left the river bed at the 'potential overflow' point marked in Figure 3. However, the related ice/rock/water mixture must have been sufficiently fast/mobile to also overflow the c. 40 m high moraine wall. The marks of this first collapse can still be identified in the very high-resolution image acquired by Quickbird on 13.1.2008 (cf. Figure 1b).

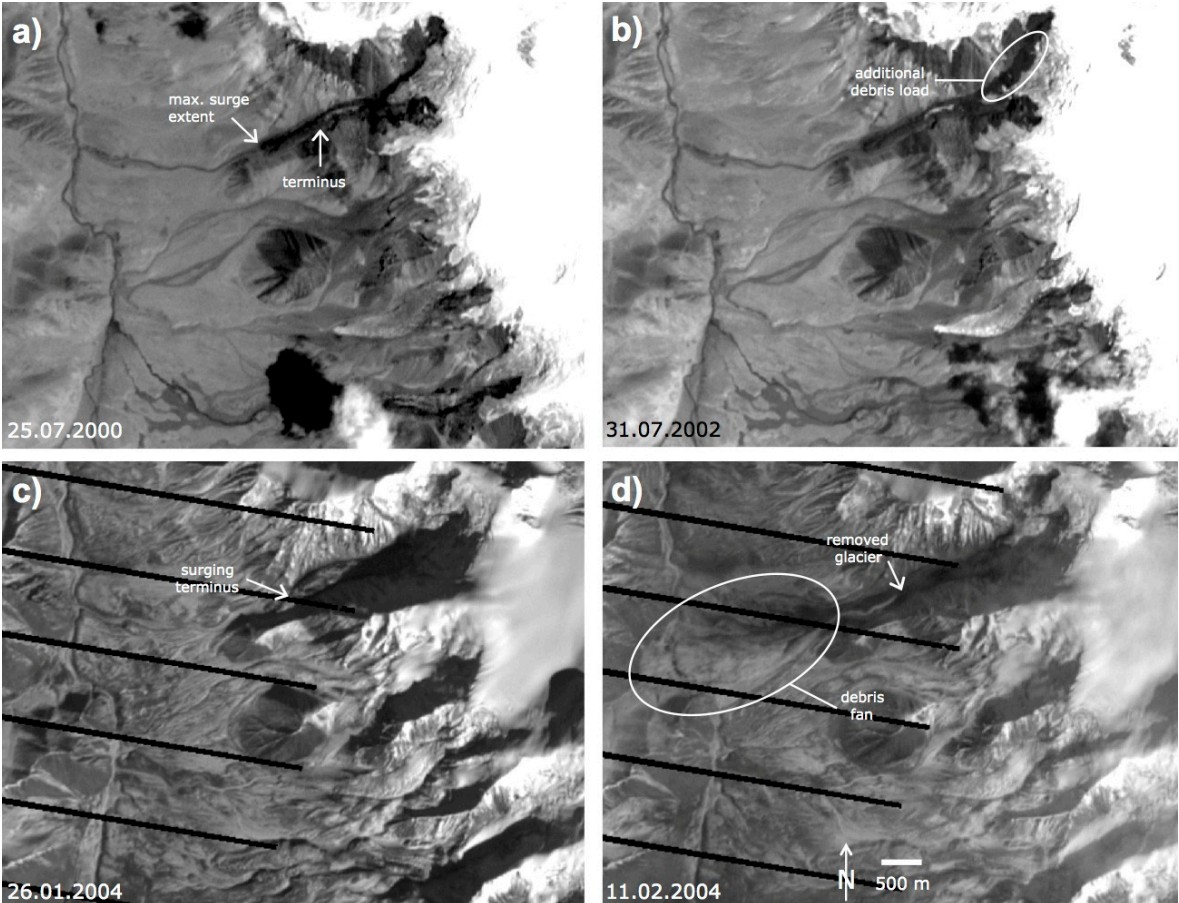

**Figure 5.** Landsat ETM+ pan images from (**a**) 25 July 2000, (**b**) 31 July 2002, (**c**) 26 January 2004, and (**d**) 11 February 2004 showing in (**a**) the active terminus in 2000 compared to the maximum surge extent, (**b**) the loading with debris by 2002, in (**c**) the maximum surge extent in 2004 before the collapse, and in (**d**) the debris fan after the collapse. Image source: earthexplorer.usgs.gov.

The full extent of the deposit has been mapped using the 15 m panchromatic band of the Landsat ETM+ scene acquired on 5 August 2004 (Figure A2a). With a measured area of 2.16 km$^2$ and an arbitrarily assumed mean thickness of maybe 10 m, the volume of the avalanche would be 20–25 million m$^3$, which is much less than both avalanches in the Aru mountain range [21]. The Landsat image also shows a lake (area 0.353 km$^2$) that likely formed as a result of the blocking of the main river by the avalanche deposit (Figure A2a). Until 14 September 2004 the lake grew to 0.534 km$^2$ and drained sometime between 29 June and 15 July 2005 at an even slightly larger size of 0.71 km$^2$.

*4.5. The 2007 Surge and Collapse*

From 2004 to September 2007, the glacier advanced by about 300 m before it collapsed again at some time between 23 September and 2 November 2007 (Figure 6). Its maximum extent before the collapse was much smaller than in 2004, close to its 'normal' minimum extent (cf. Figure A2a,b). This indicates that the surge/collapse mechanism might have been different, that lubrication by melt water might have played a role, or that the glacier has reached a different point of dynamic stability. It is well possible that after three years, the collapsed material was more a loose ice/rock mélange rather than a homogenous glacier. The contrast-stretched close-up of the Quickbird scene acquired on 13 January 2008 (Figure 7) reveals that a small part of the flat, lower glacier remained in its bed. This part is separated from highly crevassed dirty ice higher up that is still connected to the southern tributary. In other words, the rupture did not follow a potential failure zone, but occurred somewhere across the glacier.

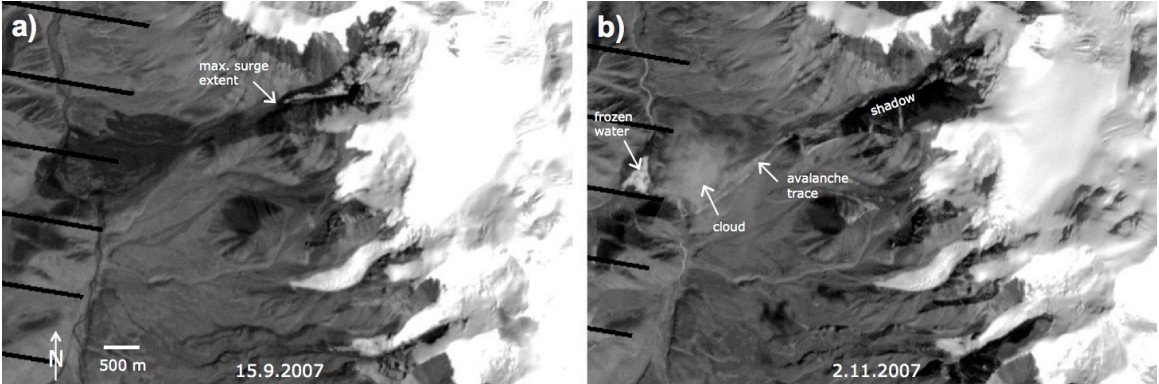

**Figure 6.** Two Landsat 7 ETM+ pan images showing the surging glacier in 2007: (**a**) before and (**b**) after the collapse. Image source: earthexplorer.usgs.gov.

The resulting avalanche deposit is well visible on a Landsat ETM+ image from 16 August 2008 (Figure A2b) that has been used for mapping its extent (1.46 km$^2$). The very low reflectance in the ETM+ panchromatic band (that stretches well into the near infrared) indicates that the deposit has a high water content, likely from the melting ice inside. The area covered by the deposit is considerably (32%) smaller than in 2004, but a part of the ice–debris mixture flowed again over the double moraine, indicating high flow velocities. The oblique perspective view in Figure 1b shows the fresh deposit on top of the older, more extended deposit from the 2004 collapse. The nadir-view of the new deposit (Figure 8) reveals several further details. It shows the larger extent of the 2004 collapse (that has flattened out) with the new 2007 deposit on top of it. The latter has still considerable relief, indicating high ice content. At the southern margin of the deposit, even crevasses can be seen, indicating that most of the former glacier might be located here. The deposit is also filling a large part of the trench and after a sharp turn to the south some material has been moved out of the trench and deposited higher up. An additional stream on top of the main deposit (with a different surface pattern) indicates that possibly a second wave of material came down after the main collapse. The image also shows the buried country road and the overtopped moraine wall.

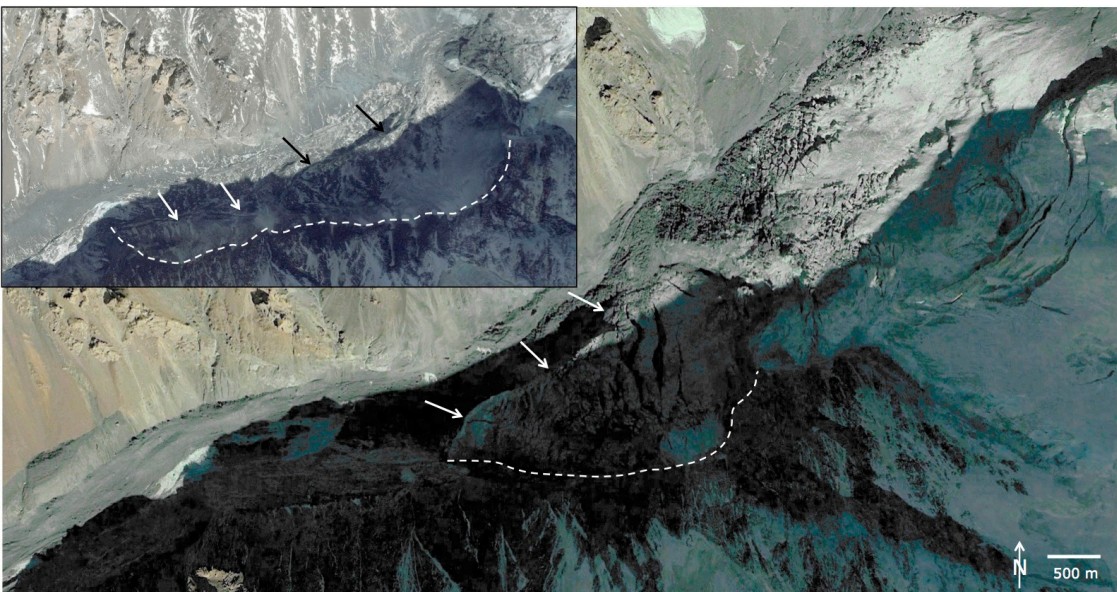

**Figure 7.** The large image shows a contrast-enhanced close-up of the glacier remnant (between the arrows and the dashed line) after its 2007 collapse as seen on the 13 January 2008 Quickbird image (image source: Screenshot from Google Earth). The smaller inset shows about the same region after the 2016 collapse on an undated satellite image. Image source: Screenshot from bing.com.

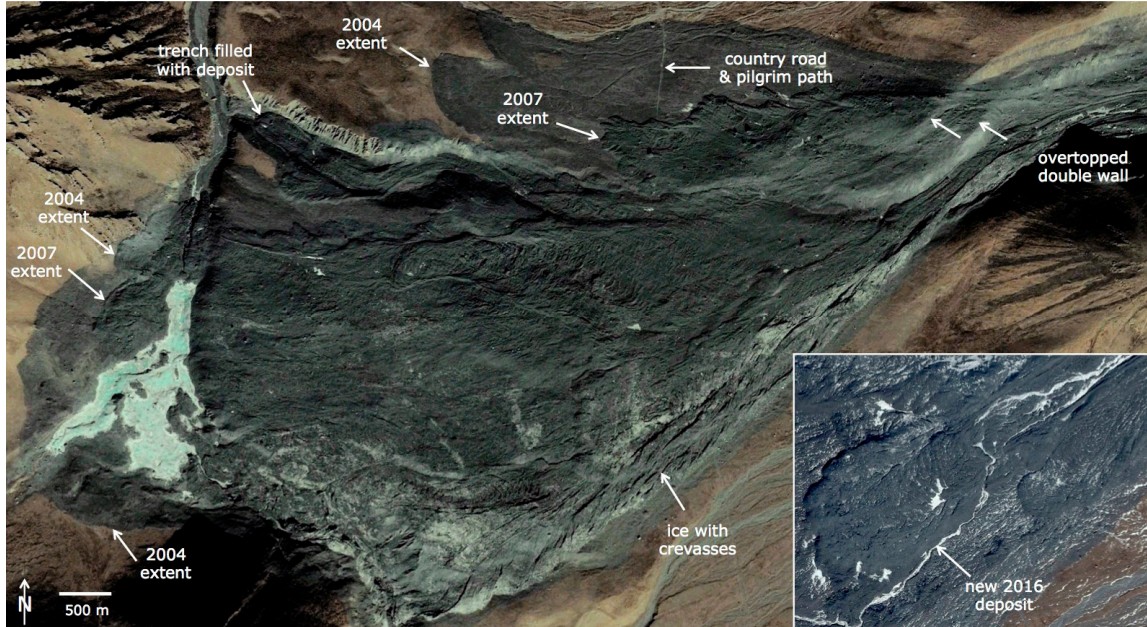

**Figure 8.** Close-up showing the avalanche deposit zone as seen on a Quickbird scene acquired on 13 January 2008, i.e., about 3 months after the 2007 collapse. Remnants from the 2004 collapse can be seen as well. Image source: Screenshot Google Earth. The inset shows a close-up of the third deposit from the 2016 collapse. Image source: Screenshot from bing.com.

### 4.6. Further Head Wall Degradation after 2009

From 2008 to 2009, another larger part of the steep glacier around the second rock outcrop (RO-B) started separating. The beginning of it is already visible on the Quickbird image from 13 January 2008 (Figure 1b). At some point between August 2010 and April 2011, this glacier part collapsed as well and likely ended on the lower glacier tongue. Interestingly, the failure zone did not follow the major crevasses but was located higher up. This means that the further development of the glaciers on this rock wall is difficult to predict. In the 15 years from 1997 to 2012, the original rock outcrop RO-A developed into a large, heart-shaped region of steep glacier-free terrain with only a small remnant of the former eastern glacier tributary left.

### 4.7. The Surge and Collapse in 2016

From 2011 to 2016, the glacier surged again by about 700 m (Figure 9) and the ice-free region around RO-B extended further upwards. The last phase of the surge and following collapse can be followed on Sentinel-2 images at 10 m spatial resolution (Figure 10). The glacier was still surging on 30 July 2016 (Figure 10a), reaching its maximum extent shortly after 28 September 2016 (Figure 10b), and collapsing until 18 October 2016 (Figure 10c). The inset in Figure 7 shows that approximately the same part of the glacier remained in its bed as after the 2007 collapse. Whereas the big gap to its upper eastern part is still there, this part is much less crevassed than in 2008.

A photograph that was taken on 12 September 2016 by G. Butler (Figure A1b) shows the advancing tongue about three weeks before its collapse. As one can see from the image, the terminus looks dual-layered with an upper layer of brighter ice over the 'normal' tongue. The image also shows some vegetation growth on top of the 2007 avalanche deposit (in the foreground), the fine grained material covering the mountain flanks to the north and south of the glacier valley (middle ground), and the steep and dark, now nearly ice-free rock wall in the background. The path through the avalanche deposit shown in Figure A1a reveals that the rocky material is rather small-grained and that the ice underneath the debris layer melted away in the meantime.

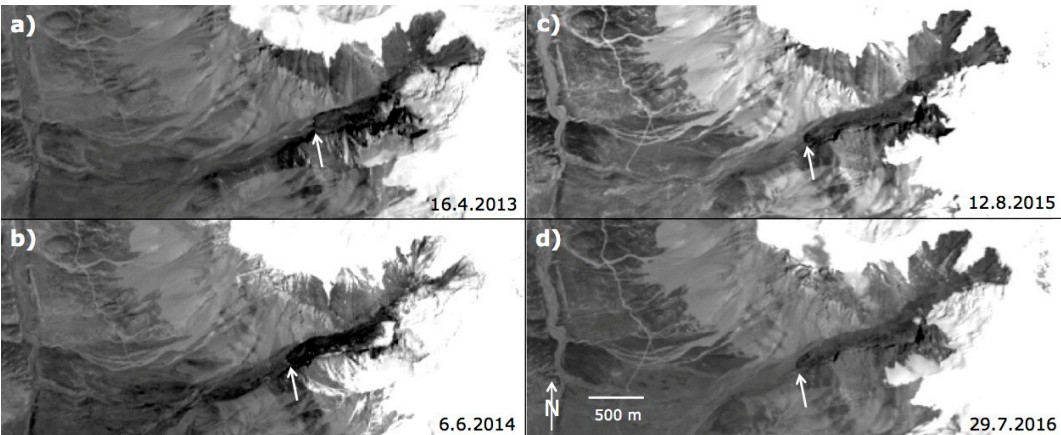

**Figure 9.** Time series of the 2013–2016 surge as seen with the Landsat 8 OLI panchromatic band. The white arrow marks the position of the terminus. Image acquisition dates are: (**a**) 16 April 2013, (**b**) 6 June 2014, (**c**) 12 August 2015, and (**d**) 29 July 2016. Image source: earthexplorer.usgs.gov.

The Sentinel-2 image from 4 August 2017 shows the deposit in nadir-view (Figure 10d) and was used to digitize its extent. It is smaller (area 1.25 km$^2$) than the one from 2007 and the avalanche has this time seemingly not overtopped the moraine walls at the end of the short valley. Hence, all material was diverted to the south-west and the deposit covers a larger region here than in 2007; very similar to 2004 (Figure 11). However, it reached not that far down, i.e. did likely not cross the main river. The inset in Figure 8 shows a subset of the fresh deposits on top of the former ones.

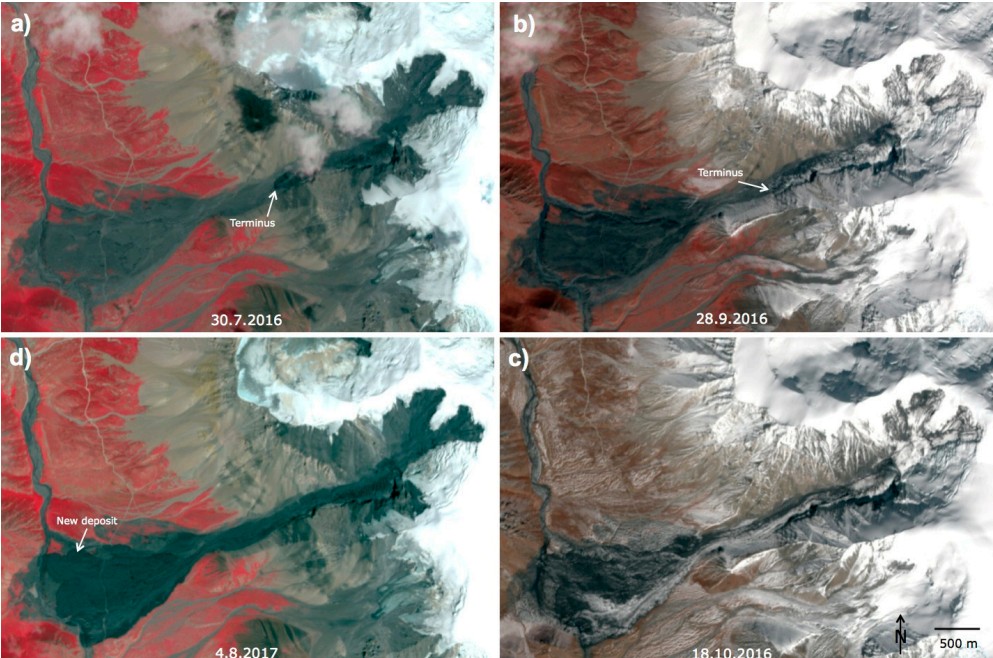

**Figure 10.** Time series of the 2016 glacier collapse as seen on false-colour composite images (near infrared, red, and green as RGB) acquired by Sentinel-2. (**a**) The image from 30 July 2016 shows the glacier during its surge. (**b**) Maximum extent on 28 September 2016 about 1 week before the collapse. (**c**) After the collapse on 18 October 2016. (**d**) The deposit from the last collapse (in darker shades of grey) in summer 2017. The destroyed country road (that is also used as a pilgrim path) has been re-established over the fresh deposit. Image source: Copernicus Sentinel data 2016 and 2017.

In Figure 11, the Sentinel-2 image from 4 August 2017 is shown along with overlays of all three deposit extents and the various glacier extents (minimum and maximum) for comparison. Remarkable are the large maximum surge extents from 2004 and 2016 compared to the extent before the 2007 collapse. The growth of rock outcrop RO-B from 2007 to 2017 and the re-established path crossing the fresh avalanche deposit (Figure A1a) can also be seen.

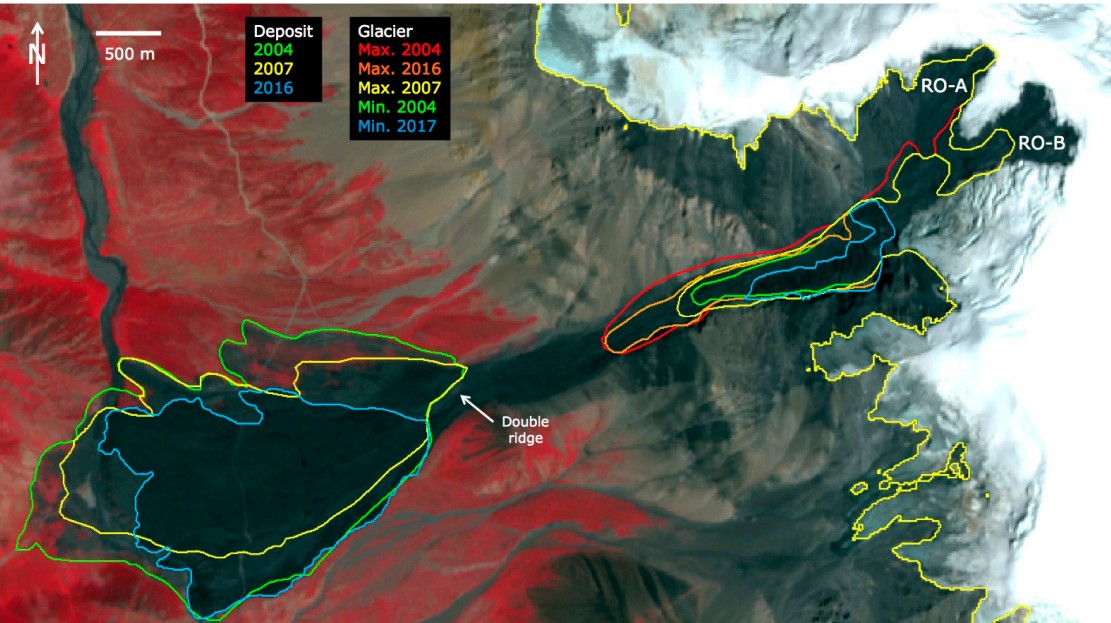

**Figure 11.** Extents of the deposits and glaciers for the three surges shown on the background of Figure 6d: Deposits from the 2004, 2007, and 2016 collapses are marked green, yellow, and blue, respectively. Maximum glacier extents (before the collapse) from 2004, 2007, and 2016 are shown in red, yellow, and orange, whereas minimum extents (after the collapse) from 2004 and 2017 are shown in green and blue, respectively. Image source: Copernicus Sentinel data 2017.

The close-up from the very high-resolution satellite image in Figure 12 (acquired after the third collapse) reveals that the bedrock is covered with fine-grained material and that the rock cliffs point downwards. There is thus not much resistance offered by the terrain and a hanging glacier that might get lubricated at its bed will have a severe stability problem. Furthermore, larger parts of the rock outcrops seem to be covered by thin ice and the rocks as well as the fine-grained material are comparably dark (i.e. they absorb much energy). It is thus well possible that the remaining glacier on this steep slope will degrade further and maybe slide down, thereby providing additional material to the lower glacier tongue. This could trigger further glacier surges and collapses in the future. Further Landsat 8 and Sentinel-2 images from 2017 and 2018 indicate that the glacier started advancing again in autumn 2017 but was in a stable position over most of 2018 at about the maximum 2007 extent shown in Figure 11.

*4.8. Elevation Changes and Topographic Analysis*

Elevation changes between the SRTM DEM and the HMA DEM over the 2000–2015 period are depicted in Figure 13 showing the difference grid and elevation profiles along a centerline through the deposit and the glacier tongue for both DEMs. Apart from regions with strong elevation gain (blue) and loss (red) that are related to DEM artifacts in regions of steep terrain and low contrast (snow, shadow), elevation changes over glaciers are also well recognizable. When starting at the top, the strong ice loss in the region of the former tributaries and later rock outcrops A and B (cf. Figure 12) is clearly visible (values are from about $-40$ to $-100$ m). Despite artifacts being close, the heart-shaped regions of the two rock outcrops match very well to the area of elevation loss.

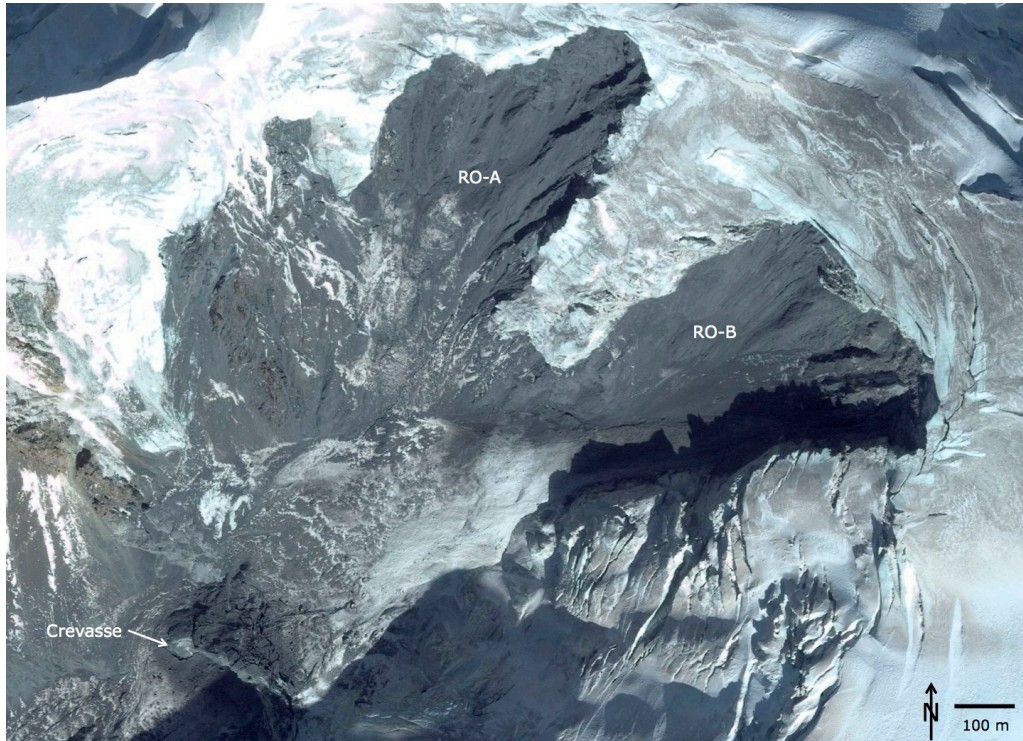

**Figure 12.** Close-up of the instable slope with the two large, heart-shaped rock outcrops (RO-A and RO-B) that have merged in the meantime. Some remaining debris-covered ice is visible (partly snow covered) in the outcrops that seem to be covered with dark and fine-grained sediment. The former eastern accumulation area is now limited to a small remaining tongue (cf. Figures 4 and A1). A larger piece of ice in the valley floor is still connected to the southern tributary but a large crevasse is separating it from the rest of the glacier. Image source: Screenshot from bing.com.

Going further down, the (former) glacier tongue is also sharply depicted in the difference image. At the acquisition date of the first DEM (SRTM), the tongue was still close to its maximum extent from the first surge (Figure 4b) that was never reached again afterwards (Figure 11). Accordingly, from 2000 to 2015, this lower region showed a pronounced thickness loss (about 40 m). When the second DEM was acquired in 2015, the glacier was again surging (Figure 9c) and the bluish region just in front of the maximum 2007 extent in Figure 13 is depicting the slightly higher (about 5–10 m) elevation of the terminus at this position compared to the year 2000 surface. The advancing tongue can also be seen in the black profile shown in the inset of Figure 13. Higher up, the glacier surface in 2015 is generally lower than in 2000 (also up to 40 m), and in the zone of the steep cliff, elevations are about the same.

Much more subtle and within the DEM uncertainty are the elevation changes for the deposit region. However, closer inspection reveals a majority of bluish cells (elevations about 5–10 m higher) within the limits of the 2007 deposit and more variable changes outside. This would confirm that this is indeed a region of mass deposit, but the uncertainties are too high for a quantitative determination of the deposited volume.

When the study region is separated in four units A: deposition, B: transition, C: flat glacier, D: steep glacier (see bottom of Figure 13), the elevation, length, and slope values presented in Table 1 are found from the HMA DEM. The mean slope values indicate that the area of deposition is indeed comparably flat (<6°) and that the flat tongue of the glacier is in the typical range for valley glaciers (about 13°). The transition zone is somewhat steeper and the mean slope of the rock walls is 36°, However, locally, values exceeding 50° are found. The overall slope or Fahrböschung (measured from the upper point of the collapsed glacier to the lowest point of the deposit) is 10.9° and thus within the range of similar events [18].

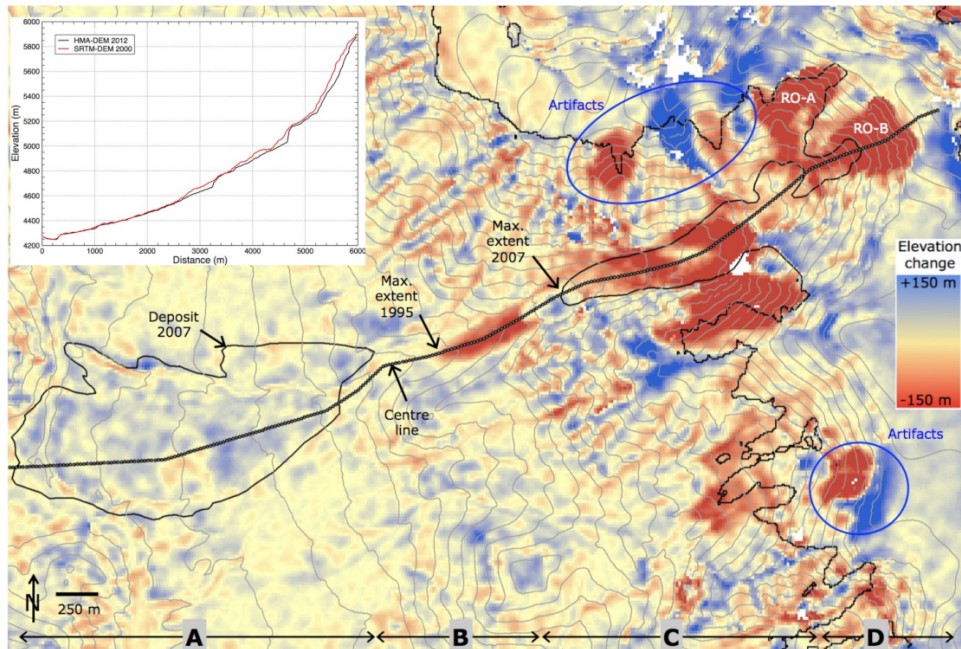

**Figure 13.** Elevation difference grid (SRTM-HMA DEM) over the 2000 to 2015 period showing volume loss (red) of the glacier but also artefacts to the north and south of it. Glacier outlines and the extent of the avalanche deposit from 2007 are shown in black, the elevation values for the inset are sampled along the dotted line. Elevation contour lines (grey) have 50 m equidistance, sections A, B, C, and D are described in the text. Elevation differences have been limited to a range of ±150 m. The inset shows elevation profiles extracted from both DEMs along the center line.

**Table 1.** Topographic characteristics of the different zones as derived from the HMA-DEM.

| | Zone Name | Elevation (m) | | | Length (m) | Mean Slope (°) |
|---|---|---|---|---|---|---|
| | | Minimum | Maximum | Range | | |
| A | Deposit | 4250 | 4450 | 200 | 2000 | 5.7 |
| B | Transition | 4450 | 4800 | 350 | 1200 | 16.3 |
| C | Flat glacier | 4800 | 5250 | 450 | 2000 | 12.7 |
| D | Steep glacier | 5250 | 5900 | 650 | 900 | 35.8 |
| | Overall slope | 4250 | 5250 | 1000 | 5200 | 10.9 |

## 5. Discussion

The detailed analysis of optical satellite images allowed reconstructing the changes of a surging and collapsing glacier in eastern Tibet in much detail. Useful scenes are available for nearly every year and much denser time series could be used to constrain the timing of the collapses down to a week when images from Landsat 5 and 7 are combined. Whereas the 30 m spatial resolution of Landsat TM is sufficient to follow major changes of the glacier and the rock outcrops (Figure 4), the 15 m panchromatic bands from Landsat 7 and 8 reveal several details much better (Figures 5, 6 and 9) and have thus been used together with 10 m resolution images from Sentinel-2 (Figure 10) to follow the changes and to map the extents of the glacier and the deposits (Figure 11). There are likely uncertainties in the delineation, but these do not impact on the general interpretation of the events. Further in-depth insights are provided by the very high-resolution satellite images from Quickbird (in Google Earth) and the image of unknown source available in bing.com that show the situation after the second and third collapses, respectively. Both images reveal fine details of the glacier remnants (Figure 7) and deposits (Figure 8), as well as of the instable rock slope with its degrading hanging glaciers and incompetent lithology (Figure 12). Several further insights can be derived from these images but they would be more speculative.

Additional quantitative information about glacier thickness changes is derived from two DEMs representing the situation after the first (SRTM DEM from 2000) and during the fourth surge (HMA DEM from 2015) of the glacier. Despite large artifacts close to the glacier, its volume loss, the removed tributaries, and a small elevation gain in the region of the deposit are visible, confirming the observations from the satellite images. In particular, the visibility of the 2015 surge front in the difference image (that is also visible in the DEM hillshade of Figure 2b) confirms the high quality of the elevation data and reliability of the results.

As a further back-up, elevation differences have also been calculated with the ALOS DEM (AW3D30) that was merged from images acquired over the 2007 to 2011 period (not shown here). These fully confirm the description of events presented above. The images for the DEM have likely been acquired shortly after the second collapse (e.g., in 2008 or 2009) and show a completely removed valley glacier, an even clearer elevation gain in the deposit area, a smaller area affected by elevation loss in the region of RO-B, and that the artifacts marked in Figure 13 are due to the SRTM DEM. In contrast, the difference between the ALOS and HMA DEM show a small surface lowering over the deposit area (from ice melting), the fully re-developed glacier tongue up to its 2015 extent, only small elevation changes in the region of RO-A and the strong surface lowering over RO-B.

Whereas the glaciologic and geomorphologic observations described above are comparably robust, any reasoning about possible processes leading to the observed events is speculative. Hence, only the more general characteristics of the Kolka and Aru glacier collapses are compared in the following. The most striking difference is that the glacier observed here already collapsed three times (2004, 2007, and 2016) in only 12 years. Repeat collapses might have also occurred for Kolkaglacier as Kotlyakov et al. (2004) [17] wrote that the 1902 surge ended "in a catastrophic outburst of ice and water" similar to the event in 2002 and that "in a few minutes, the ice spread over 9 km through the valley". However, the Kolkaglacier "eruption" in 1902 occurred "at the height of a hot summer and after heavy showers" [17] seemingly with large amounts of water, whereas only small amounts of water could have been available for the 2004 mid-winter collapse reported here. The much longer time period between the two collapses (100 years) is also different. Moreover, all collapses reported here occurred during a surge with strong frontal advance. The Aru glaciers advanced only slightly before they collapsed, but the observed elevation change pattern is fully compliant with an ongoing surge [21]. In contrast to Kolkaglacier with its well-known surge history [17], both Aru glaciers had not been known as surging before [23]. On the other hand, Kolka glacier was retreating from 1984 to 2002 and basically stagnant before its 2002 collapse.

Whereas the 2002 collapse of Kolkaglacier was likely triggered by the additional load of rock/ice avalanches on its surface [19,26], the Aru collapses seem to be triggered by extreme water pressure in combination with specific properties of the glacier bed [22]. The events reported here have likely been facilitated by the accumulation of ice and debris from the instable rock wall. However, these likely occurred in small portions and more gradually, so that the additional load might have first caused a surge rather than a sudden collapse. A similar surge initiation has been observed for Lysiiglacier in the Ak-Shirak mountain range of the Tian-Shan, where the material excavated from the Kumtor mine has been dumped on its surface [32]. Based on the chronology of events and the DEM differences, it is speculated that the 2003/04 surge was triggered by the extra load of ice and rock from RO-A, whereas the growing RO-A and RO-B triggered the 2007 surge. The third event in Oct 2016 might have been triggered by material mostly excavated from the region of RO-B. Compared to the Kolka and Aru collapses, the events reported here were much smaller in volume. Chinese authorities estimated a volume of 36 million m$^3$ for the 2004 event (as written on the information board), resulting in a mean thickness of about 15 m for the deposit. They speculated that the collapse was caused by a rising snow line and freeze-thaw action.

Regarding the short time between the first and the second collapse (3 years), one has to consider that the ice mass coming down the second (and third) time was likely more a loose ice/rock mixture rather than a compact glacier. The mechanical properties for the latter two failures might thus have been different and need further investigation. More detailed studies (e.g., on ice avalanche modeling, thermal conditions, climatic trends, bedrock lithology) are also required to understand why and how the glacier has collapsed the first time, particularly when considering that the glacier did not collapse during the first surge although it was much more extended then. Some fieldwork in this region would certainly be helpful to constrain the physical properties of the glaciers and the surrounding environment.

Overall, it seems that glacier collapses are not unique events but can occur repeatedly, particularly when there is extraordinary and continuous supply of material from surrounding rock walls. As there is still some ice left to fall on the glacier and the degradation of the steep rock walls continues, the on-going but infrequent input of ice and rock will play an important role in keeping the surges (and maybe also the collapses) alive. As the rock wall and the glacier are still very active, it can be recommended to observe them more closely in the future.

## 6. Conclusions

This study presented a description of four glacier surges and three collapses of a small valley glacier in the Amney Machen mountain range of eastern Tibet that took place from 1987 to 1995 (surge only) and in 2004, 2007, and 2016. Whereas some characteristics of the events resemble similarities of glacier collapses reported earlier (Kolka, Aru), the combination found here is unique, because they already occurred three times with only a few years in-between and—despite the magnitude of the events and their impacts on infrastructure—because they have not been reported so far in the literature. The analysis of satellite images reveals that a developing rock-slope instability might be responsible for the last three surges by infrequently adding mass on the lower glacier. However, other reasons might play a role as well and more theoretical (numerical modelling) investigations as performed for the Kolka and Aru collapses might provide additional insights into the governing processes. The description of events presented here might help in analyzing related details in future studies. Further collapses of this glacier might occur, as the supply of material from the degrading rock wall is continuing. It is thus recommended to observe this glacier more closely in the future and collect some field data to constrain modelling efforts.

**Supplementary Materials:** The following are available online at http://www.mdpi.com/2072-4292/11/6/708/s1, Key images of the surges and collapses.

**Author Contributions:** All analysis and the writing of the paper have been performed by the author (F.P.).

**Funding:** This research and the APC was funded by the ESA project Glaciers_cci, grant number 4000109873/14/I-NB.

**Acknowledgments:** This study would not have been possible without the free access to Landsat and Sentinel data, digital elevation models, and very-high resolution satellite imagery visible in Google Earth and bing.com.

**Conflicts of Interest:** The author declares no conflict of interest.

## Abbreviations

| | |
|---|---|
| DEM | Digital Elevation Model |
| ESRI | Environmental Systems Research Institute |
| ETM+ | Enhanced Thematic Mapper + |
| HMA | High Mountain Asia |
| GIS | Geographic Information System |
| GLIMS | Global Land Ice Measurements from Space |
| OLI | Operational Land Imager |
| RO | Rock Outrcrop |
| SRTM | Shuttle Radar Topography Mission |
| TM | Thematic Mapper |
| UTM | Universal Transverse Mercator |

## Appendix

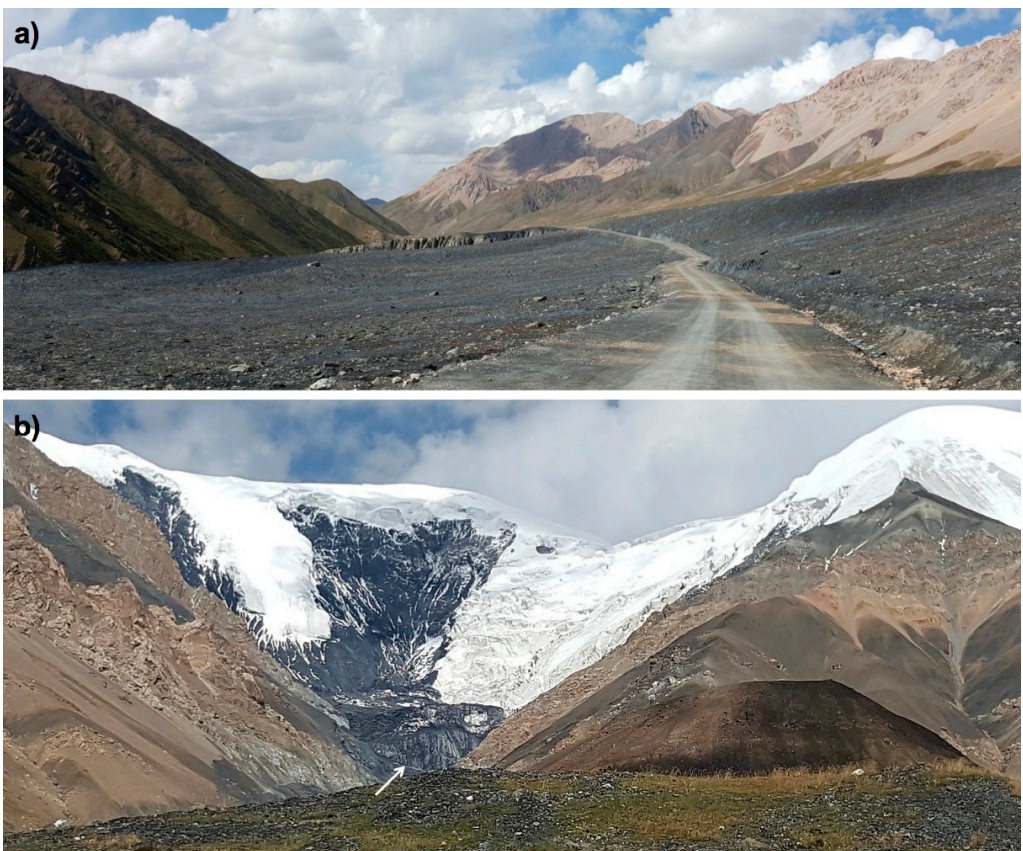

**Figure A1.** Two images from the deposit taken by Brandon G. Butler on 12 September 2016. (**a**) A view to the north showing the avalanche deposit to the left and right of the reconstructed road crossing it. Image source: [27]. (**b**) A view to the east showing the advancing glacier (the white arrow marks its front), about 3 weeks before its next collapse. Some grass is visible on the deposit of the previous collapse in the foreground. The dark rock slope in the background (to the left of the image center) is the now huge rock outcrop RO-B. Image source: [27].

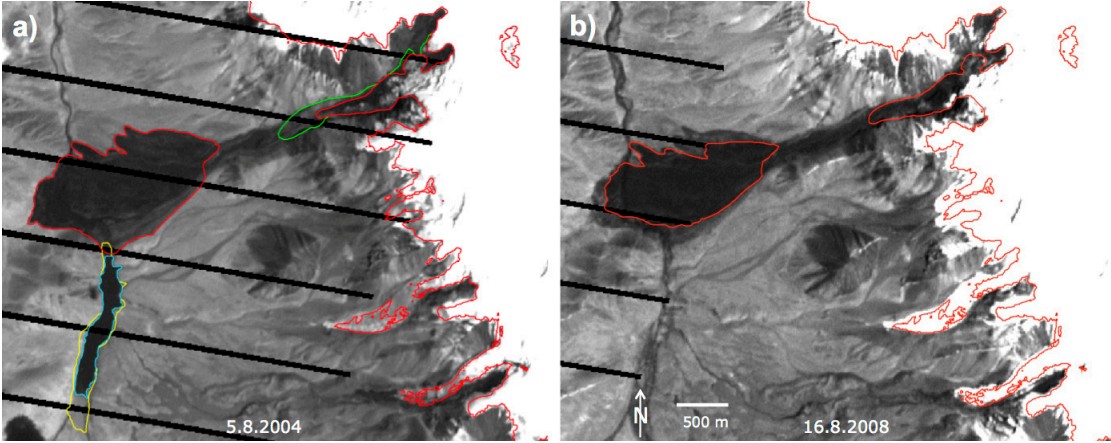

**Figure A2.** Landsat 7 ETM+ scenes (pan band) used to mark the extent of the deposits. (**a**) Scene from 5 August 2004 (green: maximum glacier extent before the 2004 collapse, red: extent of the deposit, light blue/yellow: lake extent from 5 August/14 September 2004. (**b**) Scene from 16 August 2008 (after the 2007 collapse) showing the new deposit in red. Image source: earthexplorer.usgs.gov.

**Table A1.** Overview of the satellite images used for the analysis. All Landsat scenes have path-row 133-036 and the Sentinel-2 tile used is 47SNU. Corona scenes used: #1: KH-4A scene DS1015-2164DF117, #2: KH-4B scene DS1108-2184DA110. Sensor names: L5 TM: Landsat 5 Thematic Mapper, L7 ETM+: Landsat 7 Enhanced Thematic Mapper plus, L8 OLI: Landsat 8 Operational Land Imager, S2 MSI: Sentinel-2 Multi-Spectral Instrument.

| Nr. | Sensor | Date | Nr. | Sensor | Date | Nr. | Sensor | Date |
|-----|--------|------|-----|--------|------|-----|--------|------|
| 1 | Corona | 30.12.1964 | 16 | L7 ETM+ | 16.08.2002 | 31 | L5 TM | 23.09.2007 |
| 2 | Corona | 16.12.1969 | 17 | L7 ETM+ | 03.08.2003 | 32 | L7 ETM+ | 02.11.2007 |
| 3 | L5 TM | 15.08.1987 | 18 | L5 TM | 12.09.2003 | 33 | L7 ETM+ | 16.08.2008 |
| 4 | L5 TM | 30.06.1988 | 19 | L5 TM | 14.10.2003 | 34 | L5 TM | 11.08.2009 |
| 5 | L5 TM | 21.09.1989 | 20 | L5 TM | 15.11.2003 | 35 | L5TM | 14.08.2010 |
| 6 | L5 TM | 06.07.1990 | 21 | L5 TM | 18.01.2004 | 36 | L7 ETM+ | 24.07.2011 |
| 7 | L5 TM | 11.09.1991 | 22 | L7 ETM+ | 26.01.2004 | 37 | L7 ETM+ | 12.09.2012 |
| 8 | L5 TM | 31.08.1993 | 23 | L5 TM | 03.02.2004 | 38 | L8 OLI | 16.04.2013 |
| 9 | L5 TM | 14.05.1994 | 24 | L7 ETM+ | 11.02.2004 | 39 | L8 OLI | 06.06.2014 |
| 10 | L5 TM | 20.07.1995 | 25 | L7 ETM+ | 05.08.2004 | 40 | L8 OLI | 12.08.2015 |
| 11 | L5 TM | 24.09.1996 | 26 | L5 TM | 14.09.2004 | 41 | L8 OLI | 29.07.2016 |
| 12 | L5 TM | 10.08.1997 | 27 | L7 ETM+ | 09.09.2005 | 42 | S2 MSI | 30.07.2016 |
| 13 | L5 TM | 31.07.1999 | 28 | L7 ETM+ | 26.07.2006 | 43 | S2 MSI | 28.09.2016 |
| 14 | L7 ETM+ | 25.07.2000 | 29 | L5 TM | 20.09.2006 | 44 | S2 MSI | 18.10.2016 |
| 15 | L7 ETM+ | 13.08.2001 | 30 | L7 ETM+ | 15.09.2007 | 45 | S2 MSI | 04.08.2017 |

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
