# Peer review of "Repeat Glacier Collapses and Surges in the Amney Machen Mountain Range, Tibet, Possibly Triggered by a Developing Rock-Slope Instability"

_remotesensing, doi:10.3390/rs11060708_

Round 1
Reviewer 1 Report
Dear Author, I attach the review of the manuscript.
The manuscript "Repeat glacier collapses and surges in the Amnye Machen mountain range, Tibet, possibly triggered by a developing rock-slope instability" by Frank Paul is very interesting in general and well structured.
I recommend its publication after a minor revision, the manuscript has some minor issues that have to been changed before its publication. I have all listed on the pdf.
Some general issues:
- The references in the text are all cited incorrectly. Please follow the author instructions in https://www.mdpi.com/journal/remotesensing/instructions
"In the text, reference numbers should be placed in square brackets [ ], and placed before the punctuation; for example [1], [1–3] or [1,3]. For embedded citations in the text with pagination, use both parentheses and brackets to indicate the reference number and page numbers; for example [5] (p. 10). or [6] (pp. 101–105)."
- Also, some abbreviations should be defined in parentheses the first time they appear in the abstract, main text, and in figure or table captions and used consistently thereafter.

Author Response
Please find attached my response to the comments from reviewer 1.
Reviewer 2 Report
This paper presents a unnamed glacier collapses which occurred in Eastern Tibet in 2004, 2007 and 2016 using a series of satellite optical images and DEM. The author discusses the triggering mechanism and refer to glacier instability in the future. I think this is a unique and important paper but it needs some major revisions.
My primary concern is about identification of glacier extent and some features on the glacier (Figure 3-8). The author shows a variety of optical images to track the history of the surge and collapse. Although the author uses many marks to easily understand the location of the features, I could not know the most features are true or not probably due to debris and/or image resolution. I understand such identification needs the skill and experiences, and it may be better to show the examples. At least, the description of the method for such identification needs to be added in detail for readers to understand easily how the author recognizes such features.
Here are my other comments. I hope this will help improve the manuscript.
Major comments
1: It is better to mention the definition of collapse. What condition in mountain glaciers indicates collapse? What is different from surge?
2: You use SRTM and HMA DEM to calculate elevation change. Why did you resample the both DEM to 16 m resolution? (not to 30 m resolution for SRTM)
Minor comments
P13L386: SRTM
P13L393: AW3D30
Author Response
Please find attached my response to the comments from reviewer 2.
Reviewer 3 Report
The author’ goal seems to report the repeat glacier collapses and surges in Tibet Plateau by using satellite data. This work is significant for surveying glacier collapse and surge. This manuscript is generally well-written. Thus, I have to suggest that it could be released.
Amnye Machen->A'nyemmaqen
Line 113 please check the HMA DEM sources (CT or AT)
Line 239 I would suggest that “Pilgrim path” could be changed to “country road (with function of pilgrim path for residents” .
Line 370 optical satellite imageries
Author Response
Please find attached my response to the comments from reviewer 3.
Round 2
Reviewer 2 Report
Dear Author,
The revised manuscript has been improved and I think It is acceptable for publication.
Good work!